# Haptic and Force Feedback Technology in Dental Education: A Bibliometric Analysis

**DOI:** 10.3390/ijerph20021318

**Published:** 2023-01-11

**Authors:** Min-Hsun Hsu, Yu-Chao Chang

**Affiliations:** 1School of Dentistry, Chung Shan Medical University, Taichung 40201, Taiwan; 2Department of Dentistry, Chung Shan Medical University Hospital, Taichung 40201, Taiwan

**Keywords:** haptic, haptic simulator, force feedback, virtual reality, dentistry, bibliometric analysis, dental education

## Abstract

The haptic and force feedback technology has received an increasing attention in dental schools due to its effectiveness in psychomotor skill training. However, the bibliometric analysis on haptic and force feedback technology in dental education is still scarce. Therefore, the aim of this study was to perform a bibliometric analysis of the development of haptic and force feedback technology and its changing trends in dental education. From 1 January 2001 to 30 November 2022, all papers published on haptic and force feedback technology were searched from the Web of Science Core Collection database. These data were then entered into Apple Numbers for descriptive bibliometric analysis and visualized using VOSviewer software. A total of 85 articles were retrieved following the inclusive and exclusive criteria. The results demonstrated that USA and China exhibited the most publications. The combination of correspondence author and author co-citation analysis identified the more prominent authors in this research field. The top-cited and the average citation count per year ranking led to different views of popularity. A significant increase in the number of haptic and force feedback technology publications were found in the last two years. Virtual reality is the main keyword that indicates more new integrative applications currently underway. Taken together, this study provides a detailed bibliographic analysis of haptic and force feedback technology in dental education to indicate representative authors, literatures, keywords, and trends. These detailed data will help researchers, teachers, and dental students as a very useful information when trying to make haptic and force feedback technology more prevalent in dental education in the near further.

## 1. Introduction

Haptic and force-feedback technology allows a user to experience tactile feeling or a sense of touch when operating an instrument [1]. The force-feedback device first used in dental education was developed by the University of Iowa for the detection of caries cavities [2]. Haptic and force feedback technology was considered an add-on function of the pure-vision virtual environment that could facilitate dental students with a more real operating scenario [1]. Dental students can improve their skills in haptic and force-feedback environments and let the skill levels be differentiated and identified [3]. The skill perception from haptic and force feedback technology can also be transferred to real operation situations [4].

Bibliometrics is a commonly used mathematical and statistical tool for the literature analysis of books, papers, or other publications. It is a systematic research analysis method for citation patterns in the literature. The quantitative evaluation was used to analyze bibliographic information including number of citations, citation density, citation ranking, and impact factor. In addition, the bibliometrics information about author such as co-authorship’s country, institutional affiliation, and productivity of researchers were also evaluated. These analyses are helpful for understanding the history of academic activities, identifying current research interests, and guiding potential future research directions [5]. Bibliometric analysis has been evaluated in various fields of dentistry, oral surgery and medicine such as the characteristics of a specific journal [6,7], dental education [8], and platelet-rich fibrin [5]. Since 2001, there were several articles reported on the popularity of various haptic and force feedback technology in dental education [1,9]. As far as we know, there is still limited information about the applications, demographic distributions, and the overall trend of developments about this innovative technology. Therefore, the aim of this study was to provide a bibliometric analysis of haptic and force feedback technology, trying to provide an overview and provide a different perspective on its application in dental education.

## 2. Materials and Methods

### 2.1. Database and Search Strategy Selection

The Web of Science database (Clarivate Analytics, Philadelphia, PA, USA) is the oldest database designed for more detailed citation analysis for researchers [10]. In this study, Web of Science, with its sub-databases including Science Citation Index Expanded (SCIE) and the Social Science Citation Index (SSCI), was assessed from 1 January 2001, to 30 November 2022. The search strategy was Topic = (“dental haptic”) OR Topic = (“dental force feedback”) AND Topic = (“dental education”) searched in the title and abstract sections. The exclusion criteria were applied to the following conditions: (1) not focus on haptic/force feedback technologies in dental education; (2) duplicated article; (3) a conference abstract. After the process of exclusion, the bibliometric data of the included references were exported from Web of Science for further analysis. As illustrated in Figure 1, the selective strategy was according to PRISMA flow diagram [11].

### 2.2. Bibliometric Analysis

The extracted bibliometric data with different indexes such as number of articles, authors, affiliations, publication year, citation counts, geographical distribution, keywords, and references were exported to Numbers (Apple Inc., Cupertino, CA, USA) for the descriptive bibliometric analysis. The geographical distribution was recorded from the address of correspondence author. The VOSviewer (Visualization of Similarities Viewer, VOS) software version 1.6.18 was used in this study, which is a metric analysis tool developed by Need Jan van Eck and Ludo Waltman of the Leiden University’s Centre for Science and Technology Studies in Netherland. It is a popular tool in literature metrics research to present the visualized knowledge graph of the data obtained according to the principles of literature analysis [12].

## 3. Results

### 3.1. Article Types

A total of 118 articles were identified by the search strategy from the Web of Science database. A total of 33 articles were excluded for the reasons due to not focus on haptic and force feedback technologies in dental education. The most frequently published article type was original articles (n = 75), followed by review (n = 7), editorial material (n = 2), and letter to editor (n = 1). The total 85 articles within the selected criteria were listed in Table 1.

### 3.2. Authors, Affiliations, and Country

Regarding the authorship productivity, the most frequently published corresponding authors were Dang-Xiao Wang (n = 5), Siriwan Suebnukarn (n = 5), and Yu-Chao Chang (n = 3). As shown in Figure 2, the author co-citation network contributed with a minimum of ten citations of an author were illustrated by VOSviewer. Siriwan Suebnukarn ranked as the top one scholar in the author co-citation network. About the relation to the authors’ affiliations were illustrated in Figure 3. From VOSviewer, the network visualization of thirteen co-authorship of affiliations published at a minimum of three articles were demonstrated. The top three affiliations were Beihang University, Thammasat University, and Peking University.

The most top publications from total 31 countries were USA (n = 18, 21%) and China (n = 18, 21%), followed by England (n = 9, 10.5%), and Thailand (n = 8, 9.4%). The co-authorship network of countries that contributed minimum three articles with overlay visualization were illustrated in Figure 4.

### 3.3. Citation Count

Figure 5 displays the annual number of publication per year with the annual total citation count, the number of publication increased in recent two years. The article from Thomas et al. [2] in 2001 was the first report about haptic and force feedback technology in dental education. However, there was no publication about haptic and force feedback technology in dentistry until 2006. The highest total citation counts were found in 2011.

The ranking of 30 most top-cited articles is enumerated in Table 2, the characteristics of each article was recorded as followings: article title, article type, the address of correspondence author country, publication year, total citation counts, and average citations per year. The most cited article is a review article “A review of the use of simulation in dental education” [45]. The total citation number was up to 98 during past 7 years. The second most cited article is also a review article “A review of simulators with haptic devices for medical training” [1]. The total citation number was up to 96 during past 6 years. The third most cited article is an original article “Assessment of faculty perception of content validity of PerioSim©, a haptic-3D virtual reality dental training simulator” [15]. The total citation number was up to 74 during past 15 years.

The ranking of the top 10 most average citations per year articles is enumerated in Table 3. The most cited articles per year is an original article “The application of virtual reality and augmented reality in Oral & Maxillofacial Surgery” [66] published in 2019 with 22.67 citations/year. The second most cited articles per year is an original article “A cross-sectional multicenter survey on the future of dental education in the era of COVID-19: Alternatives and implications” [73] published in 2021 with 20.00 citations/year. The third most cited articles per year is also the second most cited article shown in Table 2 “A review of simulators with haptic devices for medical training” [1] published in 2016 with 16.00 citations/year.

### 3.4. Keywords

As shown in Figure 6, a network visualization map of the most frequent author keywords (two or more common keywords) was illustrated by VOSviewer. Virtual reality is the subject of research shown in the yellow cluster. Dental education exhibits the main keyword in purple cluster associated with tooth drilling, COVID-19, and e-learning. The red cluster represented the haptic rendering domain. The brown cluster is studies related to the simulation of dental occlusion. In summary, virtual reality, dental education, and haptics were the top three keywords presented in this bibliometric analysis.

Figure 7 is portrayed by different colors within VOSviewer. The results indicate the keyword trends in recent research on haptic and force feedback technology in dental education, such as COVID-19, Simodont^®^, restorative dentistry, preclinical skill, haptic simulators, as well as tooth drilling, 3D printing, cephalometry, and dental occlusion.

## 4. Discussion

This study provides a constructive compilation of information on haptic and force feedback technology in dental education. It could help researchers to understand current trends, facilitate research for this technology, and even be implemented into regular dental curriculum. Virtual reality is the main keyword that accounts for the largest proportion of this bibliometric analysis in dental education [78]. During the current COVID-19 pandemic, virtual reality simulators have led to the development and application of this technology in many dental schools [80,93]. Haptic and force feedback technology combined with a VR simulator can compensate for the limitations of traditional phantom head-based motor skill training. In addition, unlimited reproducibility, objective evaluation of preparation by computer assessment, and cost reduction were also revealed [78]. Using haptic and force-feedback technology, it could narrow down the gap between preclinical and clinical skill level [70], reduce anxiety among novice dentists, and improve patient safety [84]. In 2015, the United Nations announced the Sustainable Development Goals, which have become a common task for all dental schools over the years. Therefore, the virtual digital 3D simulator has become a new favorite because of its ability to provide high quality education, reduce inequality, and decrease waste [87].

Currently, the application of haptic and force-feedback technology in dentistry is still limited to some disciplines, such as operative dentistry, which requires the skill of tooth preparation. In terms of recent keyword trends and the average number of citations per year, it can be found that some new techniques are being integrated with haptic and force feedback technology in new devices that have the potential to become more applicable in current preclinical learning and training. Recently, some haptic and force feedback devices could not only provide pre-clinical skill training, but also equipped with customized treatment model to achieve the goal of precision medicine. For example, intraoral scanner assembled digital models from patients were successfully imported into haptic 3D virtual reality simulator for the learning of teeth preparation before clinical managements [70,84]. Moreover, virtual 3D tooth creation form patients combined with haptic and force feedback technology were implemented in endodontic learning and training [94,95]. Recently, the haptic feedback technology was initiative in oral and maxillary surgery learning for lower impaction third molar extraction [96].

The data from WoS were only used for the analysis of the first author, resulting in the omission of other outstanding authors in the same article. In addition, the clustering results show that an author can only belong to the same group, and when the author is a cross-disciplinary or multi-specialty author, the clustering results do not naturally present the author characteristics; therefore, a modified analysis method may be developed to improve this imperfection [97].

Publications on haptic and force feedback technology in dental education were retrieved from WOS and the data was analyzed objectively and comprehensively. However, there are some limitations in this study. First, only one database Web of Science was adopted. Articles written in books, or conference proceedings were not covered in Web of Science. More databases would be added to compare the differences in the future. Second, the majority of Web of Science articles are in English. Therefore, most of the non-English language articles were neglected or excluded. Third, the recent 2-year growth trends predict an increase in the number of publications on haptic and force-feedback technology in dental education published in the preprint online database were not enrolled. Finally, the number of citations might reflect the impact or influence the article. However, this method has a potential lack of in-depth analysis of each article.

Despite these limitations, we believe the data that presented in this study still provide significant insight into the scope and type of the large body of haptic and force feedback technology in dental education. The number of censored studies of haptic and force feedback technology from WOS database over the past 20 years is relatively small (n = 85), perhaps haptic and force feedback technology is still needed to improve for mimicking more clinically relevant virtual realistic simulation training environment. However, it has many benefits in pre-clinical skills learning such as motor skill development, basic manual dexterity training and even facilitate patient safety [9,60]. Further experiments are necessary to expand and prove the training effectiveness of haptic and force feedback technology [9,80]. It will take time to make this issue more popular.

## 5. Conclusions

To the best of our knowledge, this is the first study to analyze the bibliometric data of haptic and force feedback technology in dental education. According to the results of bibliometric analysis, there has been an increasing tendency toward this topic over the most recent two years. Teaching motor skill is important for dental students. It is believed that more studies will be conducted in the future to confirm the reliability and validity of haptic and force feedback technology in dental education.

## Figures and Tables

**Figure 1 ijerph-20-01318-f001:**
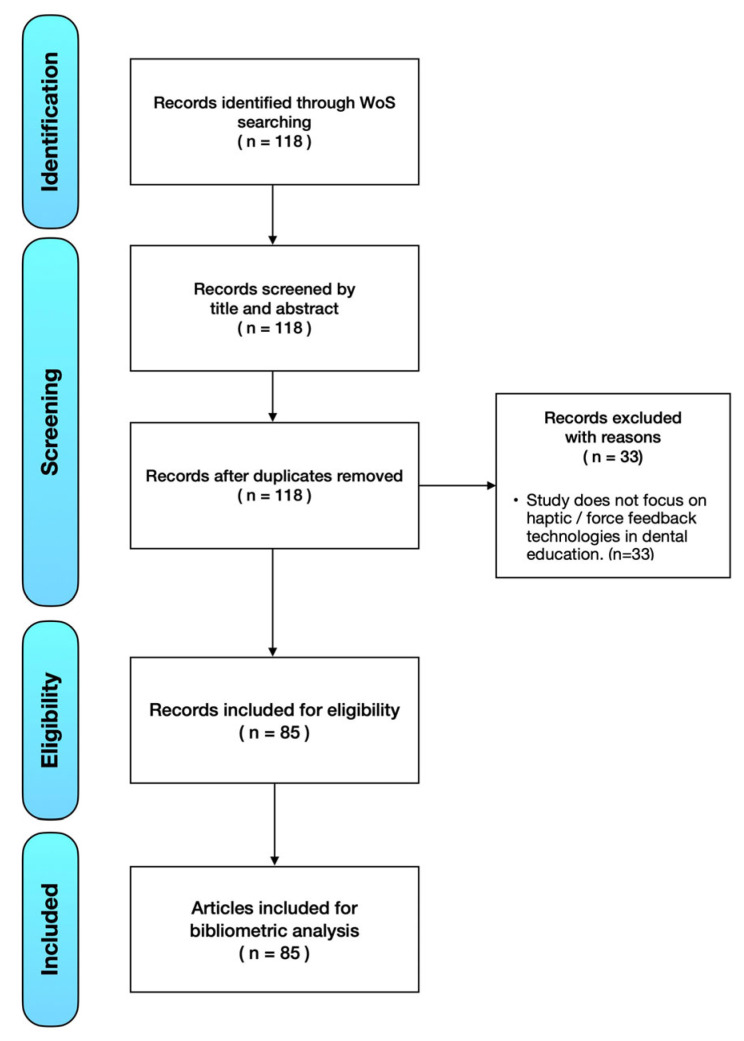
The PRISMA selective strategy flow diagram.

**Figure 2 ijerph-20-01318-f002:**
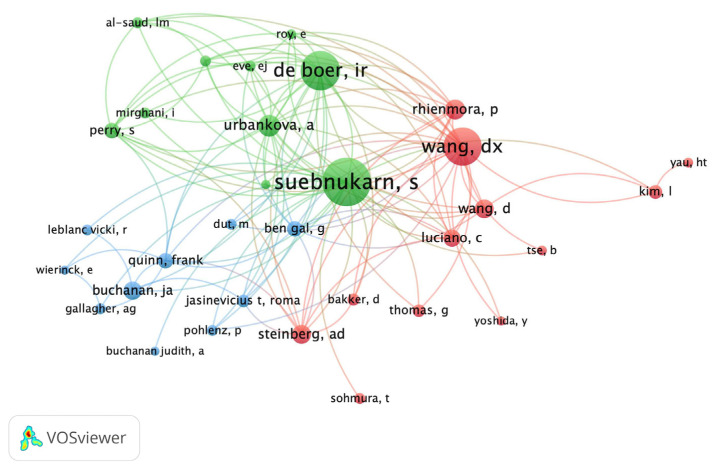
The author co-citation network that contributed with minimum ten times of citations of an author.

**Figure 3 ijerph-20-01318-f003:**
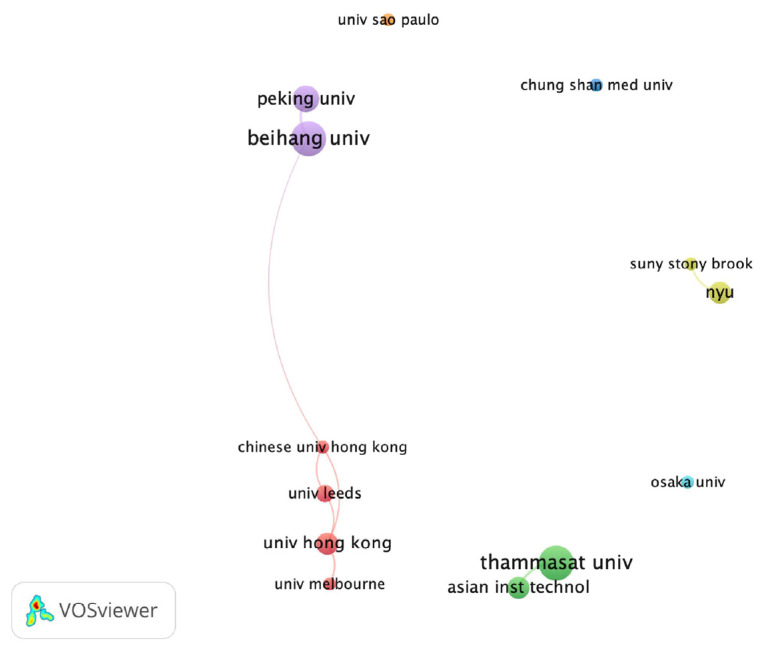
The co-authorship network of affiliations published at a minimum of three articles in network visualization.

**Figure 4 ijerph-20-01318-f004:**
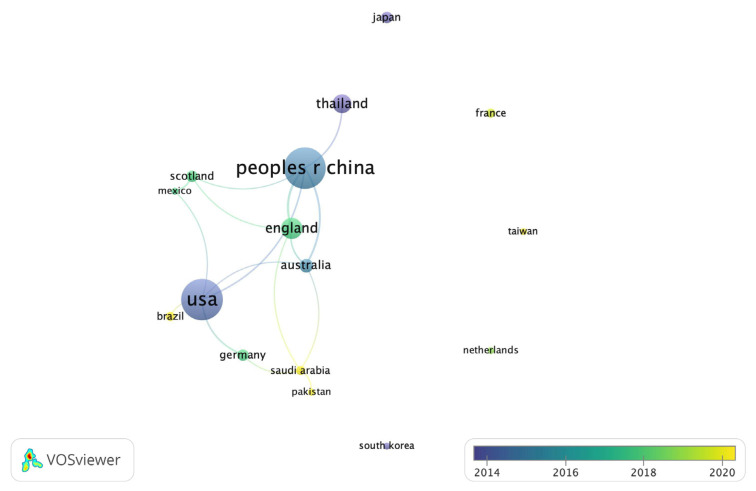
The co-authorship network of countries published at a minimum of three articles in overlay visualization.

**Figure 5 ijerph-20-01318-f005:**
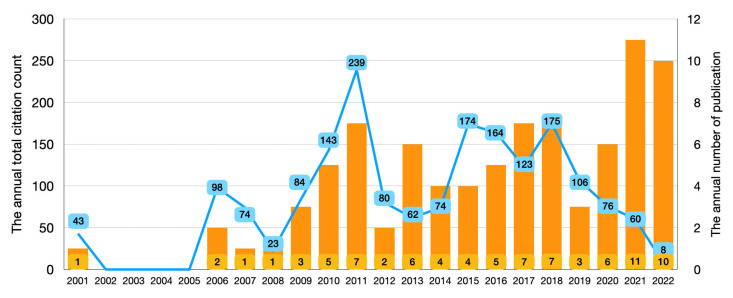
The annual total citation counts and the production of haptic and force feedback technology studies in dental education from 2001 to 2022.

**Figure 6 ijerph-20-01318-f006:**
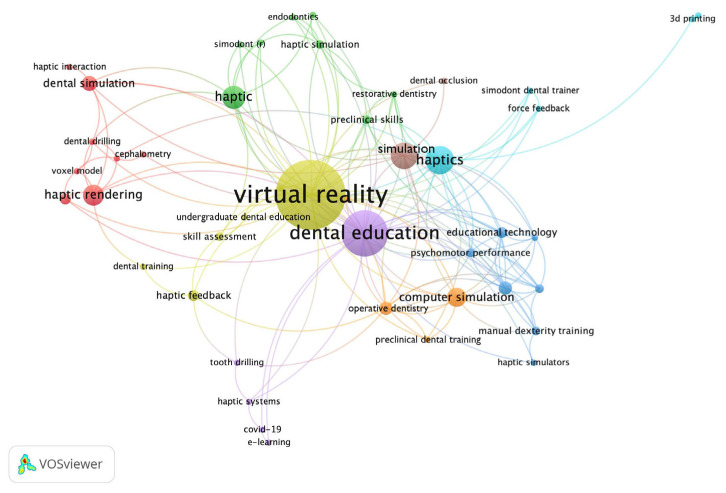
The most presented keywords of haptic and force feedback technology studies in this bibliometric analysis.

**Figure 7 ijerph-20-01318-f007:**
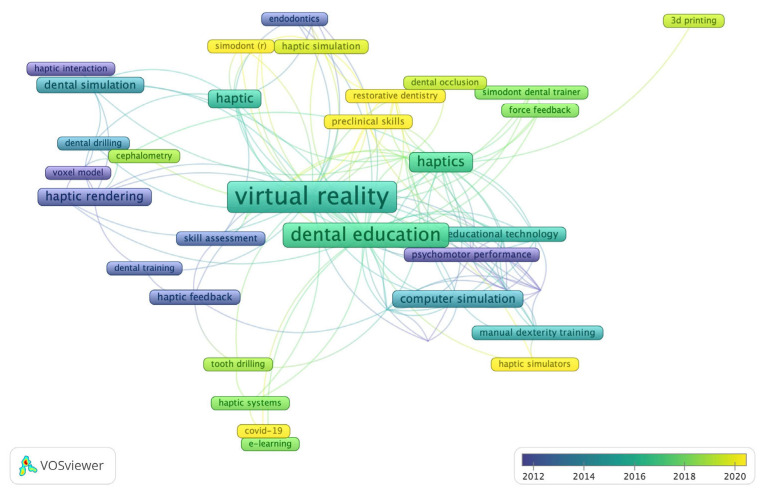
Yearly distribution of overlay author keyword visualization after normalization. It shows that more research such as tooth drilling, 3D printing, cephalometry, and dental occlusion were published recently.

**Table 1 ijerph-20-01318-t001:** The total 85 articles included in this study.

Article Number	Article Title	Journal
1	The design and testing of a force feedback dental simulator [2]	Computer Methods and Programs in Biomedicine
2	Application of virtual reality force feedback haptic device for oral implant surgery [13]	Clinical Oral Implants Research
3	Haptic interaction and volume modeling techniques for realistic dental simulation [14]	The Visual Computer
4	Assessment of faculty perception of content validity of PerioSim©, a haptic-3D virtual reality dental training simulator [15]	Journal of Dental Education
5	A virtual system for cavity preparation in endodontics [16]	Journal of Dental Education
6	Process and outcome measures of expert/novice performance on a haptic virtual reality system [17]	Journal of Dentistry
7	Application of haptic device to implant dentistry-accuracy verification of drilling into a pig bone [18]	Dental Materials Journal
8	Haptic rendering for dental training system [19]	Science in China Series F-Information Sciences
9	A virtual reality simulator for teaching and evaluating dental procedures [20]	Methods of Information in Medicine
10	Augmented kinematic feedback from haptic virtual reality for dental skill acquisition [21]	Journal of Dental Education
11	Development of a Visio-Haptic integrated dental training simulation system [22]	Journal of Dental Education
12	Toward stable and realistic haptic interaction for tooth preparation simulation [23]	Journal of Computing & Information Science in Engineering
13	Virtual dental surgery as a new educational tool in dental school [24]	Journal of Cranio-Maxillofacial Surgery
14	Access cavity preparation training using haptic virtual reality and microcomputed tomography tooth models [25]	International Endodontic Journal
15	Development of a multi-layered virtual tooth model for the haptic dental training system [26]	Dental Materials Journal
16	Intelligent dental training simulator with objective skill assessment and feedback [27]	Artificial Intelligence in Medicine
17	Passive haptic interface with MR-brakes for dental implant surgery [28]	Presence: Teleoperators & Virtual Environments
18	Physics-based haptic simulation of bone machining [29]	IEEE Transactions on Haptics
19	Preliminary assessment of faculty and student perception of a haptic virtual reality simulator for training dental manual dexterity [30]	Journal of Dental Education
20	The use of haptics to predict preclinic operative dentistry performance and perceptual ability [31]	Journal of Dental Education
21	iDental: a haptic-based dental simulator and its preliminary user evaluation [32]	IEEE Transactions on Haptics
22	Telerobotic-assisted bone-drilling system using bilateral control with feed operation scaling and cutting force scaling [33]	International Journal of Medical Robotics and Computer Assisted Surgery
23	A complex haptic exercise to predict preclinical operative dentistry performance: a retrospective study [34]	Journal of Dental Education
24	Development of a surface-based virtual dental sculpting simulator with multimodal feedback [35]	International Journal of Precision Engineering and Manufacturing
25	Graphic processing units (GPUs)-based haptic simulator for dental implant surgery [36]	Journal of Computing & Information Science in Engineering
26	Real-time medical visualization of human head and neck anatomy and its applications for dental training and simulation [37]	Current Medical Imaging Reviews
27	Testing manual dexterity using a virtual reality simulator: reliability and validity [38]	European Journal of Dental Education
28	Egocentric versus allocentric spatial ability in dentistry and haptic virtual reality training [39]	Applied Cognitive Psychology
29	Construct validity and expert benchmarking of the haptic virtual reality dental simulator [40]	Journal of Dental Education
30	Haptic simulation of organ deformation and hybrid contacts in dental operations [41]	IEEE Transactions on Haptics
31	Performance of dental students versus prosthodontics residents on a 3D immersive haptic simulator [42]	Journal of Dental Education
32	Relative contribution of haptic technology to assessment and training in implantology [43]	BioMed Research International
33	A GPU-implemented physics-based haptic simulator of tooth drilling [44]	International Journal of Medical Robotics and Computer Assisted Surgery
34	A review of the use of simulation in dental education [45]	Simulation in Healthcare
35	Virtual reality for medical training: the state-of-the-art [46]	Journal of Simulation
36	Preliminary evaluation of a virtual reality dental simulation system on drilling operation [47]	Bio-Medical Materials and Engineering
37	A review of simulators with haptic devices for medical training [1]	Journal of Medical Systems
38	Survey on multisensory feedback virtual reality dental training systems [48]	European Journal of Dental Education
39	The evaluation of a novel haptic-enabled virtual reality approach for computer-aided cephalometry [49]	Computer Methods and Programs in Biomedicine
40	Six degree-of-freedom haptic simulation of probing dental caries within a narrow oral cavity [50]	IEEE Transactions on Haptics
41	A pilot study to assess the feasibility and accuracy of using haptic technology to occlude digital dental models [51]	Journal of Dentistry
42	Feedback and motor skill acquisition using a haptic dental simulator [52]	European Journal of Dental Education
43	Getting to the root of fine motor skill performance in dentistry: brain activity during dental tasks in a virtual reality haptic simulation [53]	Journal of Medical Internet Research
44	Haptic simulation framework for determining virtual dental occlusion [54]	International Journal for Computer Assisted Radiology and Surgery
45	Penalty-based haptic rendering technique on medicinal healthy dental detection [55]	Multimedia Tools and Applications
46	Simulation and curriculum design: a global survey in dental education [56]	Australian Dental Journal
47	The effect of force feedback in a virtual learning environment on the performance and satisfaction of dental students [57]	Simulation in Healthcare
48	Virtual reality simulator for dental anesthesia training in the inferior alveolar nerve block [58]	Journal of Applied Oral Science
49	3D imaging, 3D printing and 3D virtual planning in endodontics [59]	Clinical Oral Investigations
50	3D printed surgical simulation models as educational tool by maxillofacial surgeons [60]	European Journal of Dental Education
51	A patient-specific haptic drilling simulator based on virtual reality for dental implant surgery [61]	International Journal for Computer Assisted Radiology and Surgery
52	A scoring system for assessing learning progression of dental students’ clinical skills using haptic virtual workstations [62]	Journal of Dental Education
53	Capturing differences in dental training using a virtual reality simulator [3]	European Journal of Dental Education
54	Effectiveness of the multilayered caries model and visuo-tactile virtual reality simulator for minimally invasive caries removal: a randomized controlled trial [63]	Operative Dentistry
55	Haptic, physical, and web-based simulators: are they underused in maxillofacial surgery training? [64]	Journal of Oral and Maxillofacial Surgery
56	A scoping review of the use and application of virtual reality in pre-clinical dental education [65]	British Dental Journal
57	The application of virtual reality and augmented reality in Oral & Maxillofacial Surgery [66]	BMC Oral Health
58	The effect of variations in force feedback in a virtual reality environment on the performance and satisfaction of dental students [4]	Simulation in Healthcare
59	3D-printed patient individualised models vs cadaveric models in an undergraduate oral and maxillofacial surgery curriculum: comparison of student’s perceptions [67]	European Journal of Dental Education
60	Contribution of haptic simulation to analogic training environment in restorative dentistry [68]	Journal of Dental Education
61	COVID-19 era: challenges and solutions in dental education [69]	Journal of College of Physicians and Surgeons Pakistan
62	First experiences with patient-centered training in virtual reality [70]	Journal of Dental Education
63	Psychometric analysis of a measure of acceptance of new technologies (UTAUT), applied to the use of haptic virtual simulators in dental students [71]	European Journal of Dental Education
64	Simulation training for ceramic crown preparation in the dental setting using a virtual educational system [72]	European Journal of Dental Education
65	A cross-sectional multicenter survey on the future of dental education in the era of COVID-19: Alternatives and implications [73]	Journal of Dental Education
66	Augmented reality in clinical dental training and education [74]	Journal of Pakistan Medical Association
67	Bi-manual haptic-based periodontal simulation with finger support and vibrotactile feedback [75]	ACM Transactions on Multimedia Computing, Communications, and Applications
68	Formative feedback generation in a VR-based dental surgical skill training simulator [76]	Journal of Biomedical Informatics
69	Immersion and haptic feedback impacts on dental anesthesia technical skills virtual reality training [77]	Journal of Dental Education
70	Significance of haptic and virtual reality simulation (VRS) in the dental education: a review of literature [78]	Applied Sciences
71	Software testing automation of VR-based systems with haptic interfaces [79]	The Computer Journal
72	The challenge of dental education after COVID-19 pandemic-present and future innovation study design [80]	INQUIRY: The Journal of Health Care Organization, Provision, and Financing
73	The current situation and future prospects of simulators in dental education [81]	Journal of Medical Internet Research
74	The utility of haptic simulation in early restorative dental training: A scoping review [9]	Journal of Dental Education
75	Haptic-enabled virtual training in orthognathic surgery [82]	Virtual Reality
76	A complex haptic exercise to predict pre-clinic operative dentistry performance: a prospective study [83]	Journal of Dental Education
77	Clinical relevant haptic simulation learning and training in tooth preparation [84]	Journal of Dental Sciences
78	Effect of the haptic 3D virtual reality dental training simulator on assessment of tooth preparation [85]	Journal of Dental Sciences
79	Influence of practical and clinical experience on dexterity performance measured using haptic virtual reality simulator [86]	European Journal of Dental Education
80	Perspectives on the implementation of haptic virtual reality simulator into dental curriculum [87]	Journal of Dental Sciences
81	The haptic 3D virtual reality dental training simulator as a good educational tool in preclinical simulation learning [88]	Journal of Dental Sciences
82	Train strategies for haptic and 3D simulators to improve the learning process in dentistry students [89]	International Journal of Environmental Research and Public Health
83	Usability, acceptance, and educational usefulness study of a new haptic operative dentistry virtual reality simulator [90]	Computer Methods and Programs in Biomedicine
84	Virtual aids and students’ performance with haptic simulation in implantology [91]	Journal of Dental Education
85	Guided innovations: robot-assisted dental implant surgery [92]	Journal of Prosthetic Dentistry

**Table 2 ijerph-20-01318-t002:** The top 30 most-cited articles based on the total citations.

Rank	Article Title	Document Type	Country	Year	Total Citations	Average Citations Per Year
1	A review of the use of simulation in dental education [45]	Review article	China	2015	98	14.00
2	A review of simulators with haptic devices for medical training [1]	Review article	Mexico	2016	96	16.00
3	Assessment of faculty perception of content validity of PerioSim©, a haptic-3D virtual reality dental training simulator [15]	Original article	USA	2007	74	4.93
4	The application of virtual reality and augmented reality in Oral & Maxillofacial Surgery [66]	Original article	Scotland	2019	68	22.67
5	Physics-based haptic simulation of bone machining [29]	Original article	Iran	2011	54	4.91
6	Application of virtual reality force feedback haptic device for oral implant surgery [13]	Original article	Japan	2006	54	3.38
7	Intelligent dental training simulator with objective skill assessment and feedback [27]	Original article	Thailand	2011	50	4.55
8	3D printed surgical simulation models as educational tool by maxillofacial surgeons [60]	Original article	Switzerland	2018	48	12.00
9	Preliminary assessment of faculty and student perception of a haptic virtual reality simulator for training dental manual dexterity [30]	Original article	Israel	2011	48	4.36
10	Process and outcome measures of expert/novice performance on a haptic virtual reality system [17]	Original article	Thailand	2009	47	3.62
11	iDental: a haptic-based dental simulator and its preliminary user evaluation [32]	Original article	China	2012	46	4.60
12	Virtual reality for medical training: the state-of-the-art [46]	Original article	Australia	2015	45	6.43
13	Haptic interaction and volume modeling techniques for realistic dental simulation [14]	Original article	South Korea	2006	44	2.75
14	The design and testing of a force feedback dental simulator [2]	Original article	USA	2001	43	2.05
15	Capturing differences in dental training using a virtual reality simulator [3]	Original article	England	2018	42	10.50
16	Feedback and motor skill acquisition using a haptic dental simulator [52]	Original article	England	2017	41	8.20
17	Virtual dental surgery as a new educational tool in dental school [24]	Original article	Germany	2010	41	3.42
18	A scoping review of the use and application of virtual reality in pre-clinical dental education [65]	Review article	England	2019	38	12.67
19	3D imaging, 3D printing and 3D virtual planning in endodontics [59]	Review article	England	2018	37	9.25
20	Survey on multisensory feedback virtual reality dental training systems [48]	Original article	China	2016	37	6.17
21	Telerobotic-assisted bone-drilling system using bilateral control with feed operation scaling and cutting force scaling [33]	Original article	Japan	2012	34	3.40
22	Performance of dental students versus prosthodontics residents on a 3D immersive haptic simulator [42]	Original article	USA	2014	33	4.13
23	Virtual reality simulator for dental anesthesia training in the inferior alveolar nerve block [58]	Original article	Brazil	2017	30	6.00
24	Access cavity preparation training using haptic virtual reality and microcomputed tomography tooth models [25]	Original article	Thailand	2011	30	2.73
25	Development of a multi-layered virtual tooth model for the haptic dental training system [26]	Original article	Japan	2011	30	2.73
26	A virtual reality simulator for teaching and evaluating dental procedures [20]	Original article	Thailand	2010	30	2.50
27	Development of a Visio-Haptic integrated dental training simulation system [22]	Original article	Turkey	2010	28	2.33
28	Toward stable and realistic haptic interaction for tooth preparation simulation [23]	Original article	China	2010	26	2.17
29	Haptic simulation of organ deformation and hybrid contacts in dental operations [41]	Original article	China	2014	23	2.88
30	A virtual system for cavity preparation in endodontics [16]	Original article	Greece	2008	23	1.64

**Table 3 ijerph-20-01318-t003:** The top 10 most-cited articles based on the average citations pre year.

Rank	Article Title	Document Type	Country	Year	Total Citations	Average Citations Per Year
1	The application of virtual reality and augmented reality in Oral & Maxillofacial Surgery [66]	Original article	Scotland	2019	68	22.67
2	A review of simulators with haptic devices for medical training [1]	Review article	Mexico	2016	96	16.00
3	A review of the use of simulation in dental education [45]	Review article	China	2015	98	14.00
4	A scoping review of the use and application of virtual reality in pre-clinical dental education [65]	Review article	England	2019	38	12.67
5	3D printed surgical simulation models as educational tool by maxillofacial surgeons [60]	Original article	Switzerland	2018	48	12.00
6	Capturing differences in dental training using a virtual reality simulator [3]	Original article	England	2018	42	10.50
7	3D imaging, 3D printing and 3D virtual planning in endodontics [59]	Review article	England	2018	37	9.25
8	Feedback and motor skill acquisition using a haptic dental simulator [52]	Original article	England	2017	41	8.20
9	Virtual reality for medical training: the state-of-the-art [46]	Original article	Australia	2015	45	6.43
10	Survey on multisensory feedback virtual reality dental training systems [48]	Original article	China	2016	37	6.17

## Data Availability

The data presented in this study are available on request from the corresponding author.

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
