# Peer review of "Haptic and Force Feedback Technology in Dental Education: A Bibliometric Analysis"

_ijerph, 2023, doi:10.3390/ijerph20021318_

Round 1

Reviewer 1 Report

Material and Methods: It would be advisable to add inclusion criteria or to define the exclusion criteria better.

Figures two and three are vey interesting.

3.4Keywords, you should capitalize the K.

The references. All articles analyzed should be referenced. This is the biggest change.

Author Response

Material and Methods: It would be advisable to add inclusion criteria or to define the exclusion criteria better.

Ans: It has been revised according to this expert reviewer.

Figures two and three are very interesting.

Ans: Thanks for the encouragement of this expert reviewer.

3.4 Keywords, you should capitalize the K.

Ans: It has been revised accordingly.

The references. All articles analyzed should be referenced. This is the biggest change.

Ans: All articles analyzed have been referenced in the text.

Reviewer 2 Report

The manuscript: Haptic and force feedback technology in dental education: A bibliometric analysis, analyzes a current and relevant topic for the faculties of Medicine and especially for those of Dental Medicine, where practice is a key element in the learning process.

The data provided by this analysis can represent a starting point for future studies in the field, but, some aspects require clarification as follows:

1.       In the abstract it is specified that the starting date of the analysis of the articles is from November 30, 2022. It must be corrected.

2.       The introduction should be improved by providing more comprehensive data about this field and by adding new bibliographic data.

3.       In Materials and Methods section, it is specified that the end date of the article analysis is November 30, 2022. The newest and oldest article should be specified.

4.       Also, the Discussions section should be completed and correlated with more studies in the field. Perhaps adding a few aspects that refer to the limitations of the study would be of great help for the readers.

Author Response

  1. In the abstract it is specified that the starting date of the analysis of the articles is from November 30, 2022. It must be corrected.

Ans: Thanks for the comment of this expert reviewer. The bibliometric analysis was from January 1, 2001 to November 30, 2022. It has been revised in the abstract.

  1. The introduction should be improved by providing more comprehensive data about this field and by adding new bibliographic data.

Ans: Thanks for the comment of this expert reviewer, the introduction has been improved accordingly.

  1. In Materials and Methods section, it is specified that the end date of the article analysis is November 30, 2022. The newest and oldest article should be specified.

Ans: Thanks for the comment of this expert reviewer. The bibliometric analysis was from January 1, 2001 to November 30, 2022. It has been revised in the method section.

  1. Also, the Discussions section should be completed and correlated with more studies in the field. Perhaps adding a few aspects that refer to the limitations of the study would be of great help for the readers.

Ans: The discussion has been improved according to this expert reviewer’s comments.

Round 2

Reviewer 1 Report

Despite having improved a lot, the bibliography still does not add up because if they were finally left with 85 articles, all of them should be referenced. It is a very interesting topic that could be published but more in an education journal.

Author Response

Thanks for the comments for this expert reviewer. The total 85 articles that met the criteria were all referenced in this revision.

Reviewer 2 Report

The authors improved the manuscript and I am agree for publishing in the current form.

Author Response

Thanks for the encouragement of this expert reviewer.